# Rhodium(II)-catalyzed multicomponent assembly of α,α,α-trisubstituted esters via formal insertion of O–C(sp³)–C(sp²) into C–C bonds

Dan Ba[1], Si Wen[1], Qingyu Tian[1], Yanhui Chen[1], Weiwei Lv[1] & Guolin Cheng [1✉]

The direct cleavage of C(CO)−C single bonds, delivering otherwise inaccessible compounds, is a significant challenge. Although the transition metal-catalyzed insertion of functional groups into C(CO)−C bonds has been studied, strained ketone substrates or chelating assistance were commonly required. In this article, we describe a rhodium(II)-catalyzed three-component reaction of 1,3-diones, diazoesters, and *N,N*-dimethylformamide (DMF), leading to an unusual formal insertion of O–C(sp³)–C(sp²) into unstrained C(CO)−C bonds. This procedure provides a rapid entry to a gamut of otherwise inaccessible α,α,α-trisubstituted esters/amide from relatively simple substrates in a straightforward manner. 55 examples of highly decorated products demonstrate the broad functional group tolerance and substrate scope. The combination of control experiments and isotope-labeling reactions support that O, C(sp³), and C(sp²) units derive from 1,3-diones, diazoesters, and DMF, respectively.

[1] College of Materials Science & Engineering, Huaqiao University, 361021 Xiamen, China. ✉email: glcheng@hqu.edu.cn

Ketones are widely present in natural products and synthetic molecules, and are one of the most fundamental feedstocks in organic synthesis. But while there are many reactions involving the α-functionalization and transformation of carbonyl group of ketones, the selective and catalytic cleavage of C(CO)−C single bonds is still a significant challenge[1–26]. Especially, the direct insertion of functional groups into C(CO)−C single bonds, enabling the formation of otherwise inaccessible compounds, is considerably appealing[27]. For example, the traditional Baeyer–Villiger reaction[28] and Büchner–Curtius–Schlotterbeck reaction[29–32] could directly insert one-atom into C(CO)−C bonds to give esters and homologated ketones, respectively (Fig. 1a). Recently, transition metal-catalyzed chemoselective insertion of unsaturated units (alkene, allene, alkyne, ketone, and imine) into strained C(CO)−C bonds has been extensively studied by Dong[33–38], Murakami[39,40], Cramer[41,42], Martin[43], Chi[44], and Krische (Fig. 1b)[45]. However, successful transformation for the catalytic insertion of functional groups into unstrained C(CO)−C bonds is extremely rare, wherein chelating assistance is required[46]. In 2018, Bi and co-workers reported the first example that involve the formal insertion of carbenoids into acyclic C(CO)−C bonds using Ag catalyst (Fig. 1c)[47,48]. Undoubtedly, the development of multiple functional groups, particularly deriving from different molecules, and insertion into unstrained C(CO)−C bonds is of great interest from both practical synthetic applications and mechanistic investigations.

In the course of developing transition metal-catalyzed deacylative cross-coupling of 1,3-diones with carbene precursors, we unexpectedly observed the α,α,α-trisubstituted ester products by using N,N-dimethylformamide (DMF) as solvent[49–52]. Herein we report a multicomponent synthesis of α,α,α-trisubstituted esters from 1,3-diones, diazoesters, and DMF using a rhodium(II) catalyst, in which one-oxygen, one-carbon (sp³), and one carbon (sp²), deriving from 1,3-diones, diazoesters, and DMF, respectively, are inserted into C(CO)−C bonds (Fig. 1d). This process represents a method for catalytic skeletal remodeling and a complement to cut and sew strategies based on C−C bonds cleavage[27].

## Results

**Reaction development.** We commenced our study by examining the multicomponent reaction of 1,3-diphenylpropane-1,3-dione (**1a**), methyl α-phenyldiazoacetate (**2a**), and DMF in the presence of a catalytic amount of [RuCl₂(p-cymene)]₂ at ambient temperature under air atmosphere for 12 h. The three-component reaction product (**3a**) was obtained in 51% yield (Table 1, entry 1). The structure of **3a** was unambiguously verified by single-crystal X-ray diffraction. We then explored the efficiency of the reaction using different transition metal catalysts. Rh₂(OAc)₄ and AgOAc catalysts could give the desired product in 65% and 60% yields, respectively (entries 2 and 3), whereas no product was detected using Pd(OAc)₂ and Cu(OAc)₂ catalysts (entries 4 and 5). The choice of Rh salts was also critical to this reaction. Rh(III) and Rh(I) salts exhibited inferior reactivity compared to Rh(II) salt (entries 6 and 7). A similar yield of **3a** was obtained when Rh₂(esp)₂ was used instead of Rh₂(OAc)₄, and no reaction occurred without a catalyst (entries 8 and 9). Remarkably, we noticed that 4 Å MS had a significant effect on the reactivity and gave **3a** in 85% yield (entry 10). Further investigation of the loading of catalyst and additive, as well as the concentration of the reaction afforded no better results (entries 11−15). A comparative yield was observed when the reaction was carried out under nitrogen atmosphere (entry 16).

Interestingly, the carbenoid insertion (one-carbon insertion) product was not detected in any of the investigations of the reaction parameters[47,48].

**Substrate scope.** With the optimized reaction conditions in hand, we explored the substrate scope with respect to 1,3-diones. As shown in Fig. 2a, 1,3-diarylpropane-1,3-diones bearing diverse substituents (methyl, *tert*-butyl, methoxyl, halogen, and tri-fluoromethyl) on their aryl rings smoothly underwent reactions to generate the desired α,α,α-trisubstituted esters in 60−97% yields (**3b**−**p**). The reactivity of this reaction was slightly

---

### Table 1 Optimization of the transition metal-catalyzed C(CO)−C bonds insertion reaction (reaction conditions: 1a (0.2 mmol), 2a (0.1 mmol), catalyst, and additive (50 mg) in DMF (0.5 mL) at room temperature under air atmosphere for 12 h).

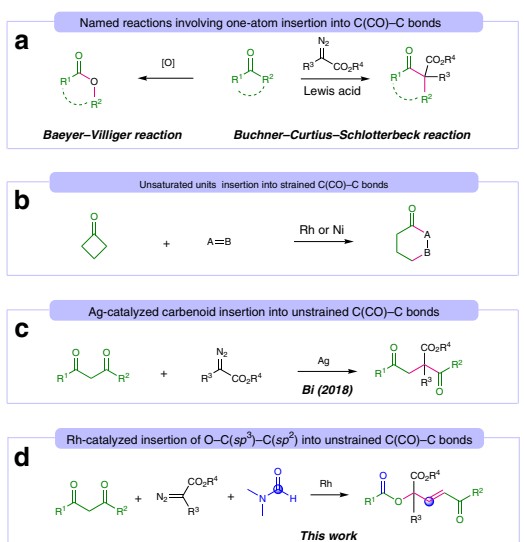

| Entry | Catalyst (x mol %) | Additive | Yield (%)ᵃ |
|---|---|---|---|
| 1 | [RuCl₂(p-cymene)]₂ (5) | — | 51 |
| 2 | Rh₂(OAc)₄ (2) | — | 65 |
| 3 | AgOAc (10) | — | 60 |
| 4 | Pd(OAc)₂ (10) | — | 0 |
| 5 | Cu(OAc)₂ (10) | — | 0 |
| 6 | [RhCp*Cl₂]₂ (2) | — | 24 |
| 7 | [Rh(cod)Cl]₂ (2) | — | Trace |
| 8 | Rh₂(esp)₂ (2) | — | 62 |
| 9 | — | — | 0 |
| 10 | Rh₂(OAc)₄ (2) | 4 Å MS | 85 |
| 11ᵇ | Rh₂(OAc)₄ (2) | 4 Å MS | 70 |
| 12ᶜ | Rh₂(OAc)₄ (2) | 4 Å MS | 81 |
| 13 | Rh₂(OAc)₄ (1) | 4 Å MS | 67 |
| 14 | Rh₂(OAc)₄ (5) | 4 Å MS | 84 |
| 15ᵈ | Rh₂(OAc)₄ (2) | 4 Å MS | 80 |
| 16ᵉ | Rh₂(OAc)₄ (2) | 4 Å MS | 84 |

ᵃIsolated yields based on **2a**.
ᵇ4 Å MS (25 mg) was used.
ᶜ4 Å MS (75 mg) was used.
ᵈ1 mL of DMF was used.
ᵉReaction was carried out under nitrogen atmosphere. esp: α,α,α',α'-Tetramethyl-1,3-benzenedipropionic acid.

---

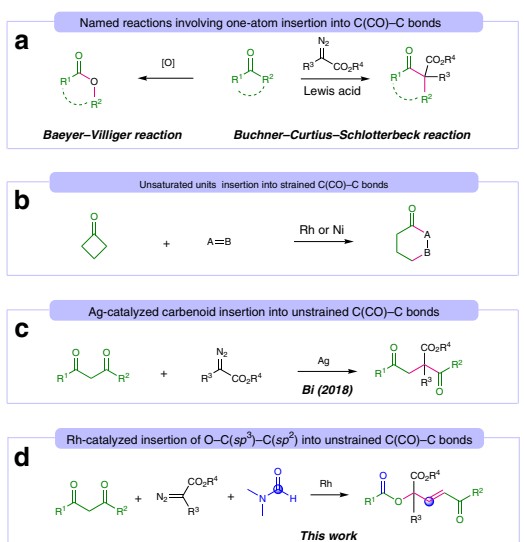

**Fig. 1 Functional groups insertion into C(CO)−C bonds. a** Named reactions involving one-atom insertion into C(CO)−C bonds. **b** Unsaturated units insertion into strained C(CO)−C bonds. **c** Ag-catalyzed carbenoid insertion into unstrained C(CO)−C bonds. **d** Rh-catalyzed insertion of O-C(sp³)-C(Sp²) into unstrained C(CO)−C bonds.

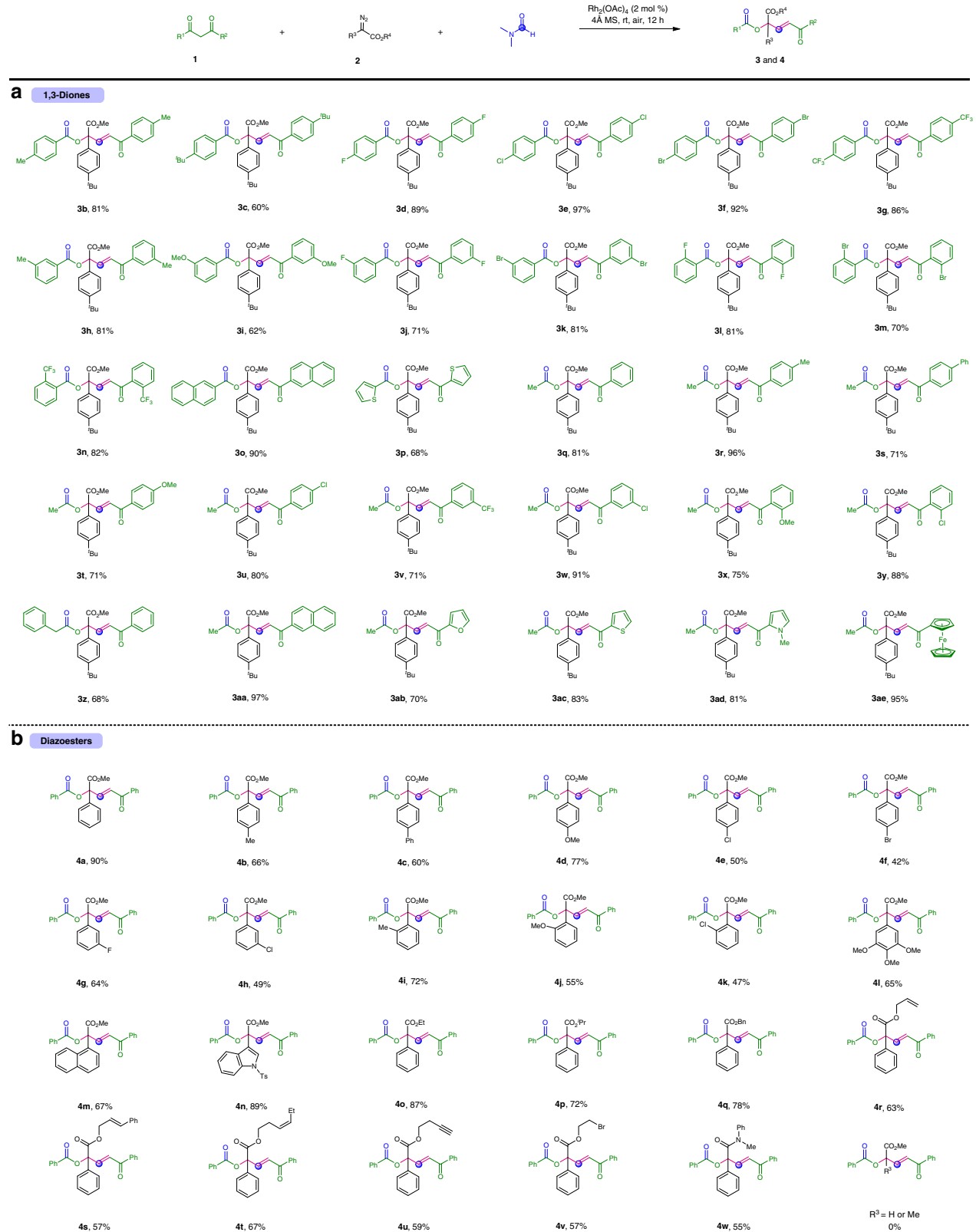

**Fig. 2 Scope of substrates. a** Scope of 1,3-diones. **b** Scope of diazoesters. Reaction conditions: **1** (0.2 mmol), **2** (0.1 mmol), Rh$_2$(OAc)$_4$ (2 mol%), and 4 Å MS (50 mg), in DMF (0.5 mL) at room temperature under air atmosphere for 12 h. Isolated yields based on **2** are shown.

influenced by the electronic properties of the aryl rings. 1,3-dia-rylpropane-1,3-diones with electron-donating groups (**3b** and **3c**) afforded lower yields of the corresponding products than those with electron-withdrawing groups (**3d**−**g**). Importantly, the steric

hindrance of the aryl rings was not observed to affect the reaction efficiency, and high yields were obtained when substrates with an *ortho*-substitutent on phenyl rings were used (**3l**−**n**). Substrates containing naphthalene and thiophene rings can be successfully

converted to the desired products (**3o** and **3p**). Remarkably, aryl alkyl 1,3-diketones delivered the desired products in good to excellent yields with exclusive chemoselectivity, and the C(CO)−C bonds cleavage reactions were found to occur selectively at the alkanoyl−carbon bonds (**3q−z** and **3aa−ae**). It is worth mentioning that the reaction is amenable to a wide range of heteroaryl-substituted 1,3-diones (**3ab−ad**). Finally, a ferrocene unit could be incorporated into the product with high efficiency (**3ae**).

Subsequently, we examined the substrate scope of α-aryldiazoacetates (Fig. 2b). We first set out to investigate the effect of substituents on the phenyl rings of α-phenyldiazoacetates. A variety of substituted methyl α-phenyldiazoacetates reacted smoothly with **1a**, furnishing the desired products (**4a−l**) in moderated to good yields. The electronic nature of the substituents on the phenyl rings had an obvious effect on the yields, wherein electron-poor groups were less favorable for this reaction (**4e−h**). In addition, the steric hindrance of the substrates had a negligible effect on the reactivity (**4i−k**). Moreover, α-naphthyl and α-indolyl diazoacetates were proved to be suitable substrates, giving the corresponding product in 67% and 89% yields, respectively (**4m** and **4n**). We then evaluated the generality of substituents on the ester moieties of α-aryldiazoacetates. Gratifyingly, ethyl, isopropyl, benzyl, allyl, cinnamyl, 3-hexenyl, homopropargyl, and 2-bromoethyl groups were well tolerated, affording the desired products (**4o−v**) in 57–87% yields. Notably, α-phenyl-$N$-methyl-$N$-phenyl diazoacetamide underwent the reaction to provide α,α,α-trisubstituted amide (**4w**) in 55% yield.

Despite the broad substrate scope shown herein (Fig. 2), this transformation is not without limitations. For example, when acetylacetone, ethyl acetoacetate, and $N$-methyl-3-oxobutanamide were subjected to the reaction, no desired products were observed. In addition, the donor−acceptor diazoesters are always required, and α-alkyl and α-H diazoesters failed to give the corresponding products. This might be explained by rapid decomposition of the carbene precursors in the presence of a rhodium(II) catalyst.

**Synthetic applications**. This three-component reaction was amenable to a gram-scale synthesis. Ester (**3a**) could be produced without modifying the standard conditions in 78% yield on a 5 mmol scale (Fig. 3). The ester and enone groups of the products of this reaction offer handles for further elaboration. To illustrate this point, several transformations of **3a** were studied. First, the 1,4-dione products (**5a** and **5b**) could be obtained via hydrolyzation/decarboxylation/isomerization cascaded reaction from **3a** and **3ae**, respectively in excellent yields, which could serve as a precursor of Paal–Knorr pyrrole synthesis. Second, treatment of **3a** with $K_2CO_3$/MeOH solution led to the transesterification/Micheal addition product (**6**) in 90% yield. Moreover, the reduction of **3a** with $NaBH_4$ proceeded efficiently to produce alcohol (**7**).

**Preliminary investigation of reaction mechanism**. To gain an insight into this Rh-catalyzed C(CO)−C bonds insertion reaction, we performed mechanistic investigations. The reactions were not inhibited by adding 2,2,6,6-tetramethyl-1-piperidinyloxy (TEMPO) or 1,1-diphenylethylene (DPE), which indicated that a radical pathway is unlikely to operate in this reaction system (Fig. 4a). The result of the reaction of unsymmetrical 1,3-dione (**1af**) and **2a** demonstrated that the chemoselectivity was slightly influenced by the electron density of aryl-groups, and the C(CO)−C bond cleavage tended to occur at the electron-poor moiety (Fig. 4b). Then, the reaction of **1a** and **2a** in

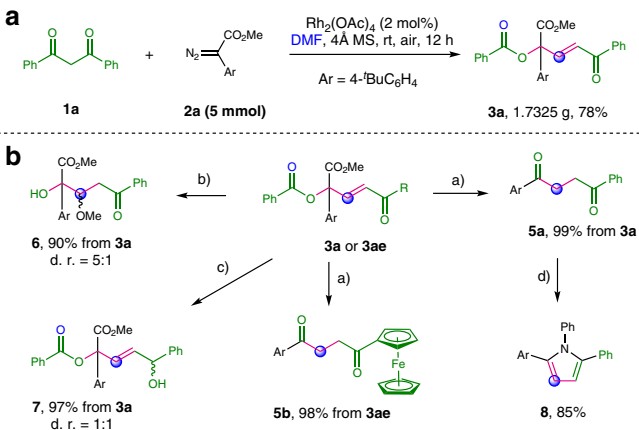

**Fig. 3 Gram-scale reaction and functional group transformations. a** Gram-scale reaction. **b** Functional group transformations. Reaction conditions: [a]**3a** (0.1 mmol), LiOH (2 equiv) in THF/$H_2O$ 4:1 (1 mL) at room temparature. [b]**3a** (0.1 mmol), $K_2CO_3$ (2 equiv) in MeOH (1 mL) at room temparature. [c]**3a** (0.1 mmol), $NaBH_4$ (1.5 equiv) in MeOH (1 mL) at room temperature. [d]**5a** (0.1 mmol), $PhNH_2$ (1.2 equiv) and TsOH (0.1 equiv) in toluene (1 mL) at 100 °C.

$N,N$-diethylformamide was carried out, leading to **3a** in 77% yield, whereas no product was formed using $N,N$-dimethylacetamide as solvent. When $N$-methyl-$N$-phenylformamide was used as solvent, **3a** and $N$-methylaniline could be isolated in 21% and 15% yields, respectively (Fig. 4c). Importantly, the reaction with DMF-formyl-$^{13}$C as solvent gave **3a-$^{13}$C** in 86% yield with 99% incorporation (Fig. 4d). The result of the reaction with DMF-dimethyl-$^{13}C_2$ indicated that the one carbon source could hardly originate from the $N$-methyl group of DMF (Fig. 4e). Furthermore, we carried out deuterium-labeling experiments using DMF-$D7$ (Fig. 4f). The full incorporation of deuterium was observed at β position of the ester product (**3a-D-1**). These results suggest that the formyl group of DMF may serve as the one-carbon source[53]. Next, when the reaction was conducted in the presence of 5 equiv of $D_2O$, the incorporation of deuterium at γ position of product (**3a-D-2**) was observed (Fig. 4g). Moreover, 5% of the oxygen atom of ester (**3q-$^{18}$O-1**) was labeled using 2 equiv of $H_2^{18}O$, and no $^{18}$O-labeled transesterification/Micheal addition product (**6**) was detected (Fig. 4h). These reactions indicate that water may be generated during the reaction process, and the benzoyl oxygen atom of **3a** originates from the in situ generated water. Finally, when the $^{18}$O-labeled 1,3-dione (**1q-$^{18}$O**) was subjected to the reaction, 22% and 12% of the oxygen atom of **3q-$^{18}$O-2** and **6-$^{18}$O** were labeled, respectively, which indicate that the oxygen atom at α position of **3q** derives from 1,3-dione (Fig. 4i).

On the basis of the control experiments and literature reports, we proposed a plausible reaction mechanism (Fig. 5). Initially, the reaction of α-aryldiazoacetate (**2**) with Rh(II) catalyst generates Rh(II) carbene complex (**A**), which is captured by enolate (**B**) to give oxonium ylide (**C**). Then, the nucleophilic addition of oxonium ylide (**C**) to DMF affords aldehyde intermediate (**D**), followed by intramolecular aldol reaction of **D**, affording the dihydrofuran intermediate (**E**). Finally, intermediate **E** could be converted into α,α,α-trisubstituted esters (**3** and **4**) via a *retro*-Baylis−Hillman-type reaction.

**Discussion**

In conclusion, we demonstrate an example of Rh(II)-catalyzed formal insertion of O−C(sp$^3$)−C(sp$^2$) into unstrained C(CO)−C

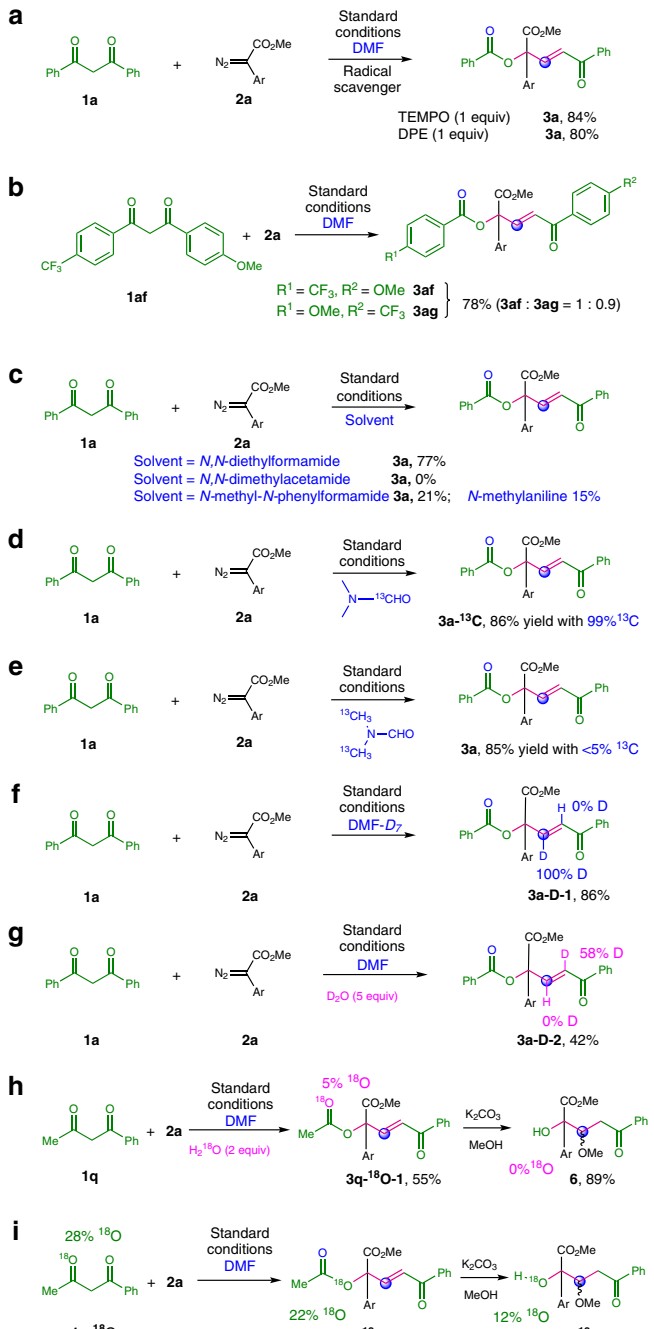

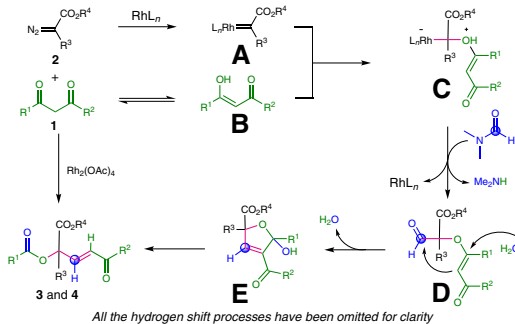

**Fig. 5 Plausible reaction mechanism.** A plausible mechanism involves the carbene insertion reaction of intermediates **A**, **B** to afford oxonium ylide **C**, subsequent nucleophilic addition, intramolecular aldol reaction, and retro-Baylis−Hillman-type reaction to form the desired products **3** and **4**.

## Methods

**General procedure for preparation of 3**. A screw-capped reaction vial was charged with 1,3-diones (**1**) (0.2 mmol, 2 equiv), methyl 2-(4-(*tert*-butyl)phenyl)-2-diazoacetate (**2a**) (23.2 mg, 0.1 mmol, 1 equiv) 4 Å MS (50 mg), and DMF (0.5 mL), followed by the addition of Rh$_2$(OAc)$_4$ (0.8 mg, 0.002 mmol, 2 mol%). The resulting mixture was stirred at room temperature for 12 h, until TLC showed the complete consumption of **2a**. After the reaction was completed, the reaction mixture was evaporated under reduced pressure to leave a crude mixture, which was purified by column chromatography on silica gel (eluting with ethyl acetate/petroleum = 1:10) to afford **3**.

## Data availability

All relevant data are available in Supplementary Information and from the authors. The X-ray crystallographic coordinates for structures reported in this study have been deposited at the Cambridge Crystallographic Data Centre (CCDC), under deposition numbers CCDC: 1975575 (**3a**). These data can be obtained free of charge from The Cambridge Crystallographic Data Centre via www.ccdc.cam.ac.uk/data_request/cif.

**Fig. 4 Control experiments and isotope-labeling reactions. a** Radical trapping experiments. **b** The reaction of unsymmetrical 1,3-dione. **c** The reactions using other amide solvents. **d** [13]C-labeled experiment using DMF-formyl-[13]C as solvent. **e** [13]C-labeled experiment using DMF-dimethyl-[13]C$_2$ as solvent. **f** Deuterium-labeling experiments using DMF-*D7* as solvent. **g** Deuterium-labeling experiments in the presence of D$_2$O. **h** [18]O-labeled experiment in the presence of H$_2$[18]O. **i** The reactions of [18]O-labeled 1,3-dione.

bonds. A preliminary mechanistic study reveals that O, C(sp$^3$), and C(sp$^2$) units originate from 1,3-diones, diazoesters, and DMF, respectively. This transformation proceeds under mild reaction conditions and opens up a versatile synthetic entry to highly decorated α,α,α-trisubstituted esters/amide from readily accessible starting materials. The development of asymmetric version of this procedure is currently under investigation in our laboratory.

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

## Acknowledgements

This work was supported by the NSF of China (21672075) and the Instrumental Analysis Center of Huaqiao University.

## Author contributions

D.B. developed the C(CO)−C bond cleavage reaction. S.W., Q.T., Y.C., and W.L. explored the substrate scope. G.C. conceived and supervised the project. G.C. wrote the manuscript.

## Competing interests

The authors declare no competing interests.
