## [Peer Review File · Nature Communications]

Reviewers' comments:

Reviewer #1 (Remarks to the Author):

Ketones are widely present in natural products and synthetic molecules, and are one of the most fundamental feedstock in organic synthesis. In this work, the authors described a Rhodium(II)-catalyzed formal insertion of O-C(sp³)-C(sp²) into C(O)-C bonds. This procedure provides a rapid entry to a gamut of otherwise inaccessible α,α,α -trisubstituted esters/amide from relatively simple substrates in a straightforward manner. Mechanism investigations demonstrated that O, C(sp³), and C(sp²) units derive from 1,2-diones, diazoesters, and DMF, respectively. This method represents a new type of metal-catalyzed cleavage of unstrained C-C bonds, and all the products were characterized well. As a whole, the manuscript is well written and this method might be useful to the wide readership of this journal. Therefore, this one would like to recommend the manuscript for publication in Nature Communications, after addressing the following concerns:

- (1) In the scope of 1,3-diones, if in 1,3-diones, one side is a phenyl group with a strong electron-withdrawing group and the other is a phenyl group with an electron-donating group, which side is more inclined to occur the cleavage of C(CO)-C bond?
- (2) In the scope of diazoesters, I think the author should explain why the donor-acceptor diazoesters are always required, and α -alkyl and α -H diazoesters failed to give the corresponding products. It will help readers understand the reaction.
- (3) In the title of Fig. 2, misspelled the word of diazoesters.
- (4) In the mechanism, did the authors detect the formation of dimethylamine in the reaction system? And in the transformation from D to E, where the water derived from?
- (5) In the acknowledgements, there are two words "and t" that are redundant.
- (6) In the references, the format is inconsistent, the references 1, 58, 59 and 60 are not justified, and the authors wrote only one author in some references, while wrote all authors in others.
- (7) In Supporting Information, since almost all products in this article are new compounds, all solid products should have a melting point test.

Reviewer #2 (Remarks to the Author):

Cheng and co-workers report a Rh(II)-catalyzed three-component reaction of 1,3-diones, diazoesters, and DMF. The reaction gave an unusual products that feature a formal insertion of O-C-O into C(CO)-C bonds. Indeed the reaction is quite unusual and the authors have also carried out control experiments to gain insights into the reaction mechanism. However, the work is not recommended for publication in Nature Communications for the following reasons.

- (1) For transition-metal-catalyzed reaction of carbene, it is quite common to observe unusual and complicated results. This is due to the diverse reactivity of the metal carbene species. So in this manuscript, although unusual reaction has been disclosed, from the reaction mechanism point of view it is still not surprising.
- (2) The reaction is quite special, the usefulness of the products are very limited. From the mechanistic point of view, the work does not show novel chemistry. The issue of the cleavage of C-C bond is actually not that much relevant to the work described in this manuscript. Overall, the work should be published in a more specialized journal.

Reviewer #3 (Remarks to the Author):

This manuscript by Cheng described a Rh(II)-catalyzed formal insertion of O-C(sp³)-C(sp²) into C(CO)-C bonds. This protocol showcased remarkable regioselectivity, broad substrate scope and good functional group tolerance, and provided an efficient pathway for the straightforward synthesis of α,α,α -trisubstituted esters from relatively simple substrates under mild condition.

In fact, this research is very interesting, but I still have some questions noted below:

1. In Fig.2, how is it going when an unsymmetric diaryl ketone was used as substrate? Besides, are aryl β -ketoesters or β -ketoamides compatible with the reaction conditions? Please conduct more investigations and provide some necessary descriptions. It should be noted if isomers were observed.
 2. For crystallographic data, there are 19 Alert level C, which could not provide enough supports, so it would be better if the data were updated and the structure of the products must be ensured carefully.
 3. According to the mechanism experiments, the HRMS spectrums of the O18-labeled products were provided. In my opinion, more detailed intensity data about the spectrums was in need to demonstrate the ratio of the isotope-labeled products. Furthermore, it's meaningful to conduct some radical trapping experiments and intermediates detection to get more information about this reaction. In addition, there is a need to cite some literatures about the reaction in that DMF serves as the one-carbon source of the C(sp²) units, and the mechanism proposed should be reasonable.
 4. The manuscript and supporting information must be checked carefully, for example, in the reference of the manuscript, the format need be checked. Additionally, ref. 52, 55, 56, 57 are not appropriate and the citation need more considerations.
- Overall, I recommend the publication of this work in Nat. Commun. after revisions.

Reviewers' comments:

Reviewer #1 (Remarks to the Author):

Ketones are widely present in natural products and synthetic molecules, and are one of the most fundamental feedstock in organic synthesis. In this work, the authors described a Rhodium(II)-catalyzed formal insertion of O-C(sp³)-C(sp²) into C(O)-C bonds. This procedure provides a rapid entry to a gamut of otherwise inaccessible α,α,α -trisubstituted esters/amide from relatively simple substrates in a straightforward manner. Mechanism investigations demonstrated that O, C(sp³), and C(sp²) units derive from 1,2-diones, diazoesters, and DMF, respectively. This method represents a new type of metal-catalyzed cleavage of unstrained C-C bonds, and all the products were characterized well. As a whole, the manuscript is well written and this method might be useful to the wide readership of this journal. Therefore, this one would like to recommend the manuscript for publication in Nature Communications, after addressing the following concerns:

(1) In the scope of 1,3-diones, if in 1,3-diones, one side is a phenyl group with a strong electron-withdrawing group and the other is a phenyl group with an electron-donating group, which side is more inclined to occur the cleavage of C(CO)-C bond?

Responses: The result of the reaction of unsymmetrical 1,3-dione (**1af**) and **2a** demonstrated that the chemoselectivity was slightly influenced by the electron density of aryl-groups, and the C(CO)–C bond cleavage tended to occur at the electron-poor moiety (Fig. 4b).

(2) In the scope of diazoesters, I think the author should explain why the donor-acceptor diazoesters are always required, and α -alkyl and α -H diazoesters failed to give the corresponding products. It will help readers understand the reaction.

Responses: This might be explained by rapid decomposition of the carbene precursors in the presence of rhodium catalyst (α -Alkyl and α -H diazoesters are more active than donor-acceptor diazoesters).

(3) In the title of Fig. 2, misspelled the word of diazoesters.

Responses: We have followed this advice and made change.

(4) In the mechanism, did the authors detect the formation of dimethylamine in the reaction system? And in the transformation from D to E, where the water derived from?

Responses: When *N*-methyl-*N*-phenylformamide was used as solvent, **3a** and *N*-methylaniline could be isolated in 21% and 15% yields, respectively (Fig. 4c). We proposed that the intramolecular aldol reaction of **D** leading to **E** generates one equivalent of water.

(5) In the acknowledgements, there are two words “and t” that are redundant.

Responses: We have followed this advice and “and t” was removed.

(6) In the references, the format is inconsistent, the references 1, 58, 59 and 60 are not justified, and the authors wrote only one author in some references, while wrote all authors in others.

Responses: We have followed this advice and removed references 1, 58, 59 and 60. When the number of authors is more than five, only the first author was shown in some references.

(7) In Supporting Information, since almost all products in this article are new compounds, all solid products should have a melting point test.

Responses: We have followed this advice and tested the melting points of all solid products.

Reviewer #2 (Remarks to the Author):

Cheng and co-workers report a Rh(II)-catalyzed three-component reaction of 1,3-diones, diazoesters, and DMF. The reaction gave an unusual products that feature a formal insertion of O-C-O into C(CO)-C bonds. Indeed the reaction is quite unusual and the authors have also carried out control experiments to gain insights into the reaction mechanism. However, the work is not recommended for publication in Nature Communications for the following reasons.

(1) For transition-metal-catalyzed reaction of carbene, it is quite common to observe unusual and complicated results. This is due to the diverse reactivity of the metal carbene species. So in this manuscript, although unusual reaction has been disclosed, from the reaction mechanism point of view it is still not surprising.

(2) The reaction is quite special, the usefulness of the products are very limited. From the mechanistic point of view, the work does not show novel chemistry. The issue of the cleavage of C-C bond is actually not that much relevant to the work described in this manuscript.

Overall, the work should be published in a more specialized journal.

Reviewer #3 (Remarks to the Author):

This manuscript by Cheng described a Rh(II)-catalyzed formal insertion of O–C(sp³)–C(sp²) into C(CO)–C bonds. This protocol showcased remarkable regioselectivity, broad substrate scope and good functional group tolerance, and provided an efficient pathway for the straightforward synthesis of α,α,α -trisubstituted esters from relatively simple substrates under mild condition.

In fact, this research is very interesting, but I still have some questions noted below:

1. In Fig.2, how is it going when an unsymmetric diaryl ketone was used as substrate? Besides, are aryl β -ketoesters or β -ketoamides compatible with the reaction conditions? Please conduct more investigations and provide some necessary descriptions. It should be noted if isomers were observed.

Responses: The result of the reaction of unsymmetrical 1,3-dione (**1af**) and **2a** demonstrated that the chemoselectivity was slightly influenced by the electron density of aryl-groups, and the C(CO)–C bond cleavage tended to occur at the electron-poor moiety(Fig. 4b). In addition, when acetylacetone, ethyl acetoacetate, and *N*-methyl-3-oxobutanamide were subjected to the reaction, no desired products were observed.

2. For crystallographic data, there are 19 Alert level C, which could not provide enough supports, so it would be better if the data were updated and the structure of the products must be ensured carefully.

Responses: For crystallographic data, there are 6 alert level C caused by incomplete of filling data (PLAT052_ALERT_1_C, PLAT053_ALERT_1_C, PLAT054_ALERT_1_C, PLAT055_ALERT_1_C, PLAT199_ALERT_1, PLAT200_ALERT_1_C). The other 13 alert level C are caused by the benzene ring and the *tert*-butyl disorder in the structural molecules. We have carried out disorder processing and imposed command restrictions on the corresponding bonds. At present, the 19 alert level C have been resolved and we also have updated the revised cif file and check cif file.

3. According to the mechanism experiments, the HRMS spectrums of the O18-labeled

products were provided. In my opinion, more detailed intensity data about the spectrums was in need to demonstrate the ratio of the isotope-labeled products. Furthermore, it's meaningful to conduct some radical trapping experiments and intermediates detection to get more information about this reaction. In addition, there is a need to cite some literatures about the reaction in that DMF serves as the one-carbon source of the C(sp²) units, and the mechanism proposed should be reasonable.

Responses:

1) We have followed this advice and more detailed intensity data about the spectrums have been shown in Supporting Information Page S32–S36.

2) The reactions were not inhibited by adding 2,2,6,6-tetramethyl-1-piperidinyloxy (TEMPO) or 1,1-diphenylethylene (DPE), which indicated that a radical pathway is unlikely to operate in this reaction system (Fig. 4a).

3) In order to obtain solid evidence that the formyl group of DMF may serve as the one-carbon source, more isotope-labeling experiments were conducted. The reaction with DMF-formyl-¹³C as solvent gave **3a**-¹³C in 86% yield with 99% incorporation (Fig. 4d). The result of the reaction with DMF-dimethyl-¹³C₂ indicated that the one carbon source could hardly originate from the *N*-methyl group of DMF (Fig. 4e). We also cited a recently review on DMF as multipurpose building block. (*Angew. Chem. Int. Ed.* **51**, 9226-9237 (2012)).

4. The manuscript and supporting information must be checked carefully, for example, in the reference of the manuscript, the format need be checked. Additionally, ref. 52, 55, 56, 57 are not appropriate and the citation need more considerations.

Responses: We have followed this advice and removed references 52, 55, 56, and 57.

Overall, I recommend the publication of this work in *Nat. Commun.* after revisions.

REVIEWERS' COMMENTS:

Reviewer #1 (Remarks to the Author):

In general the authors have responded well to the concerns raised by the reviewers. There are still some minor mistakes should be checked prior to publication.

1) There was no arrow in Fig. 1C.

2) The format need be checked carefully, for example, the title of Fig. 1d and Reaction conditions of Fig. 2.

3) In data availability, CCDC: 1975575 (3aa) should be CCDC: 1975575 (3a).

Reviewer #3 (Remarks to the Author):

The authors have carefully revised their manuscript. The scope of substrates and the mechanism have been investigated further and some experiment data and descriptions have been updated appropriately. In summary, I recommend the publication of this work in Nat. Commun. in its current form.

Dear Referees,

Thank you for your constructive comments on our manuscript. We have modified the text to address the concerns.

REVIEWERS' COMMENTS:

Reviewer #1 (Remarks to the Author):

In general the authors have responded well to the concerns raised by the reviewers. There are still some minor mistakes should be checked prior to publication.

1) There was no arrow in Fig. 1C.

Responses: We have followed this advice and made change.

2) The format need be checked carefully, for example, the title of Fig. 1d and Reaction conditions of Fig. 2.

Responses: We have followed this advice and made changes.

3) In data availability, CCDC: 1975575 (3aa) should be CCDC: 1975575 (3a).

Responses: We have followed this advice and made change.

Reviewer #3 (Remarks to the Author):

The authors have carefully revised their manuscript. The scope of substrates and the mechanism have been investigated further and some experiment data and descriptions have been updated appropriately. In summary, I recommend the publication of this work in Nat. Commun. in its current form.

Yours sincerely,

Guolin Cheng